# Bioengineered Fluorescent Nanoprobe Conjugates for Tracking Human Bone Cells: In Vitro Biocompatibility Analysis

**DOI:** 10.3390/ma14164422

**Published:** 2021-08-07

**Authors:** Christiane L. Salgado, Alexandra A. P. Mansur, Herman S. Mansur, Fernando J. M. Monteiro

**Affiliations:** 1i3S-Instituto de Investigação e Inovação em Saúde, Universidade do Porto, Rua Alfredo Allen, 208, 4200-135 Porto, Portugal; fjmont@ineb.up.pt; 2INEB, Instituto Nacional de Engenharia Biomédica, 4200-135 Porto, Portugal; 3Center of Nanoscience, Nanotechnology, and Innovation-CeNano2I, Department of Metallurgical and Materials Engineering, Federal University of Minas Gerais-UFMG, Av. Antônio Carlos, 6627, Belo Horizonte 31270-901, Brazil; alexandramansur.ufmg@gmail.com (A.A.P.M.); hmansur@demet.ufmg.br (H.S.M.); 4Department of Metallurgical and Materials Engineering, Federal University of Minas Gerais-UFMG, Av. Antônio Carlos, 6627, Belo Horizonte 31270-901, Brazil; 5FEUP, Faculdade de Engenharia, Departamento de Engenharia Metalúrgica e de Materiais, Universidade do Porto, 4200-465 Porto, Portugal

**Keywords:** quantum dots, phosphoserine signaling, MSCs differentiation, bone regeneration, caveolae endocytosis, biomaterials

## Abstract

Herein, we validated novel functionalized hybrid semiconductor bioconjugates made of fluorescent quantum dots (QD) with the surface capped by chitosan (polysaccharide) and chemically modified with O-phospho-L-serine (OPS) that are biocompatible with different human cell sources. The conjugation with a directing signaling molecule (OPS) allows preferential accumulation in human bone mesenchymal stromal cells (HBMSC). The chitosan (Chi) shell with the fluorescent CdS core was characterized by spectroscopical (UV spectrophotometry and photoluminescence), by morphological techniques (Transmission Electron Microscopy (TEM)) and showed small size (ø 2.3 nm) and a stable photoluminescence emission band. The in vitro biocompatibility results were not dependent on the polysaccharide chain length (Chi with higher and lower molecular weight) but were remarkably affected by the surface modification (Chi or Chi-OPS). In addition, the efficiency of nanoparticles uptake by the cells was dependent on cells nature (human primary cells or cell lines) and tissue source (bone or skin) in the presence or absence of the OPS modification. The complex cellular uptake pathways involved in the cell labeling with the nanoparticles do not interfere on the normal cellular biology (adhesion and proliferation), osteogenic differentiation, and gene expression. The bone cells particles uptake evaluation showed a possible pathway by Caveolin-1 that regulates cell transduction in the membrane’s Caveolae. Caveolae mediates non-specific endocytosis, and it is upregulated in HBMSC. The OPS-modified nanoparticles promoted an intense intracellular trafficking by the HBMSCs that showed late-osteoblast phenotype with an increase of extracellular matrix (ECM) mineralization (Alizarin red and Von Kossa staining for calcium phosphate crystals). In this work, the OPS modified bioconjugated QD proved to be a reliable and stable fluorescent bioprobe for cell imaging and targeting research that could also help in clarifying some cellular mechanisms of particles intracellular traffic through the cytoplasmic membrane and osteogenic differentiation induction. The in vitro HBMSC’s biocompatibility responses indicated that the OPS-modified chitosan QDs have a prospective future in laboratory and pre-clinical applications such as bioimaging analysis and for ex-vivo cellular evaluation of biomedical implants.

## 1. Introduction

Bone remodeling is an intricate process, involving the recruitment of mesenchymal stem cells (MSCs) and their differentiation into new bone under the influence of soluble signals [1]. Currently, two strategies are employed for bone regeneration: (i) localized delivery of soluble biological signals (e.g., bone metalloproteinase type 2 (BMP-2)) and (ii) a tissue engineering approach, where MSCs cultured within a scaffold and are pre-programmed in vitro using biomolecules to induce the cells differentiation into the desirable tissue [2]. In this sense, it should be interesting to use molecules that are prevalent in native bone extracellular matrix, promote osteogenesis, and hydroxyapatite nucleation [3]. One example is O-phospho-L-serine (OPS), which could be responsible for stimulating mesenchymal stem cells proliferation and inducing pre-osteoblast phenotype by upregulating their osteogenic gene expression (e.g., BMP-2, osteoponin—OPN) [4]. Particles’ surface modification with OPS could successfully mimic the biochemical structure of OPN present in osteoid formation during bone repair by enhancing alkaline phosphatase activity [5]. Consequently, there is a growing interest in developing novel fluorescent nanoparticles to specifically target bone cells (e.g., stem cells) that induce the MSC differentiation without undesirable effects in the surrounding tissues or vital organs (e.g., kidney and liver). Thus, quantum dots (QDs) that are semiconductor nanoparticles are a very promising bioimaging tool due to their unique optoelectronic properties, such as photochemical and biological stability, broad excitation, sharp emission spectra, and flexible surface properties that can be adjusted through the composition of the nanocrystals and/or the chemical structure of the ligand. QDs could be conjugated with bioactive molecules that together could target a specific cell and elucidate the dynamic process of bone regeneration, tracking MSCs and clarifying the mechanisms of cellular nanoparticles uptake, migration, proliferation and osteogenic differentiation. As a noninvasive imaging modality, the bioconjugated-QDs could be used in pre-clinical research and may be applied to living subjects with minimal animal invasiveness. The real-time bioimaging system will follow the MSC’s behavior in bone remodeling, repair, and healing following a tissue engineering strategy [6,7]. However, live imaging of cells organelles and their function with sufficient resolution has remained an unsolved drawback. Physico-chemical properties of the cell cytoplasm environment are difficult to access without changing the cellular metabolism, as it concerns a complex pathway with simultaneous intracellular cascades. In this scenario, chitosan, a polycationic natural polymer, was used to functionalize the QD-nanoparticles surface, as its properties, such as hydrophilicity, mechanical behavior, and chemical and physical stability, are directly influenced by the polymer de-acetylation degree (DD) and the molecular weight (MW) [8]. Recent studies successfully proved that it is possible to have nanoprobes with different cores (carbon, graphene, copper, etc.) functionalized by chitosan on their surface [9,10,11]. The production of CdS QDs using chitosan followed a “green-route” synthesis with an aqueous colloidal dispersion that has been recently referred [12,13]. Therefore, the CdS QD-based nanomaterial was developed as a minor nanoprobe to image complex cell pathways and basic mechanisms involved in the osteogenic differentiation of MSCs. Consequently, Cd-based QDs have been facing severe concerns such as their potential to decrease cellular viability associated with the presence of heavy metal core, that should severely restrict their applications [14,15,16]. This work aimed to develop adequate and biocompatible QDs by designing and producing biologically and environmentally safer nanomaterials [17]. In this study, a CdS QD functionalized with chitosan (Chi) was modified with O-Phospho-L-serine (OPS) that is a component of many proteins, as result of post-translational modifications [18]. OPS is an important signaling biomolecule present in native bone proteins and has been immobilized on a Collagen-nanohydroxyapatite matrix, showing an increase in MSC osteogenic differentiation, inducing an early pre-osteoblast phenotype, leading to in vitro and in vivo production of bone proteins (OPN) and precipitation of calcium phosphate crystals at the ECM after 4 and 8 weeks after scaffolds’ implantation [19,20]. OPS has also been involved in other tissue biology such as mediating partial inhibition of microglial phagocytosis [21] and influencing the immune response [22]. In addition, the effect on the biocompatibility and cell particle uptake of different Chi molecular weight (higher and lower weight) on the QD’s surface modification was studied. The final objective was to develop an in vivo bioimaging nanoparticle for monitoring and tracking the bone regenerative process at cellular level, as a pre-clinical tool for bone tissue engineering applications.

## 2. Materials and Methods 

### 2.1. Materials

O-Phospho-L-serine (OPS, (HO)_2_P(O)OCH_2_CH(NH_2_)CO_2_H), chitosan (Chi) with low molecular weight (LM, Mw = 50–190 kDa, degree of deacetylation, DD = 96.1%) and high molecular weight (HM, Mw = 310–375 kDa, DD = 78.2%), N-Ethyl-N’-[3-dimethylaminopropyl]carbodiimide hydrochloride (EDC, C_8_H_17_N_3_·HCl), N-hydroxysulfosuccinimide sodium salt (Sulfo-NHS, C_4_H_4_NNaO_6_S), ethalonamine hydrochloride (H_2_NCH_2_CH_2_OH·HCl), sodium hydroxide (NaOH), cadmium perchlorate hydrate (Cd(ClO_4_)^2^·6H_2_O) and sodium sulfide (Na_2_S∙9H_2_O) were supplied by Merck (Darmstadt, Germany). Acetic acid was supplied by Labsynth (Diadema, São Paulo, Brazil).

Chemicals were provided without any purification process. Deionized water (DI water, Millipore Simplicity^TM^, Burlington, MA, USA) with resistivity of 18 MΩ cm was used for the solutions, at room temperature (RT, 23 ± 2 °C), unless when specified otherwise.

### 2.2. Bioconjugation of the Peptides (OPS) to Chitosan (Chi)

The OPS was bioconjugated to the Chi polysaccharide backbone (Chi-OPS) as described in a previous work [23]. Briefly, chitosan was used with two Mw using EDC as a “zero-length” conjugation agent in the presence of Sulfo-NHS. EDC/sulfo-NHS converts the carboxyl groups on OPS to amine-reactive sulfo-NHS esters that react with amino groups of Chi yielding to stable covalent amide bonds.

Chitosan solution (1%, *w*/*v*) was produced as described before [23], but Chi was used with different molecular weight (LM and HM) in acetic acid at 2% and, pH was enhanced to 6 through a basic solution (NaOH—0.1 mol L^−1^). OPS conjugation of was performed with EDC solution (0.5 mmol L^−1^), S-NHS solution (1.0 mmol L^−1^) and OPS solution (20.0 mg mL^−1^) for 10 min at low temperature (6 °C). Chitosan solution was mixed at a 1.0:1.5 molar ratio (chitosan monomer: OPS) and incubated for 1.5 h (RT). Afterwards, 2-Aminoethanol hydrochloride (1.0 µM) was added to quench the reaction for 15 min.

### 2.3. Production of CdS/Polysaccharide Conjugates 

The CdS nanoparticles stabilized by chitosan (QD_Chi, LM or HM) or chitosan–O-phospho-L-serine (QD_Chi–OPS, LM or HM) were synthesized as previously described [23]. Briefly, 47 mL of Chi–OPS or Chi solutions (Chi = 0.4 mg mL^−1^) in DI water were added to the reaction vessel and the pH, when necessary, was adjusted to 6.0 ± 0.1 (NaOH, 0.1 mol L^−1^). Under moderate magnetic stirring, 4 mL of Cd^2+^ precursor solution (Cd(ClO_4_)_2_·6H_2_O, 1.0 × 10^−2^ mol L^−1^) and 2.5 mL of S^2−^ precursor solution (Na_2_S·9H_2_O, 0.8 × 10^−2^ mol L^−1^) were added to the flask and stirred for 3 min. The obtained CdS QDs dispersions were dialyzed for 24 h in a cellulose membrane (cut-off cellulose membrane, MWCO = 14,000 Da). Afterwards, the solutions were stored at 6 + 2 °C.

### 2.4. Chemic-Physical Evaluation of CdS/Polysaccharide Conjugates 

Ultraviolet–visible spectrophotometry (UV–Vis) measurements were performed in transmission mode (n = 3) using Lambda EZ-210 (Perkin-Elmer, Medtech, Waltham, MA, USA).

Photoluminescence (PL) spectra of QDs were obtained using FluoroMax-Plus-CP (Horiba Scientific, São Paulo, Brazil) at excitation = 400 nm.

Photon correlation spectroscopy or quasi-elastic light scattering and measured by the electrochemical equilibrium (ZP, ζ-potential) analyses that were done in a ZetaPlus instrument (Brookhaven Instruments Corporation, Holtsville, NY, USA) with a laser light wavelength of 660 nm (35 mW red diode laser) and using a minimum of ten replicates.

QDs images were obtained by transmission electron microscope (TEM, Tecnai G2-20-FEI, FEI Company, Hillsboro, OR, USA) at 200 kV. 

### 2.5. Biological Characterization of CdS Conjugates 

#### 2.5.1. Cell Viability Assay

For the in vitro study, a human cell line (human osteoblast-like cell line—MG63, ATCC, Manassas, VA, USA), human dermal fibroblast (HDFn, Institute Corriel, Camden, NJ, USA), and human primary MSCs (human bone stromal stem cell—HBMSC, São João Hospital, Porto, Portugal) were maintained as described earlier [21]. Briefly, the cells were cultures in α-MEM (Merck, Darmstadt, Germany) cell culture medium with 10% (*v*/*v*) FBS (Gibco, New York, NY, USA) and 1% (*v*/*v*) penicillin–streptomycin (100 U/mL PEN, 100 mg/mL STREP, Gibco, Life Technologies, Carlsbad, OR, USA). The primary cells were kept in 5% CO_2_ atmosphere at 37 °C until the third passage (P3). The samples toxic effect on cells was evaluated using a standard Alamar blue assay (Rezasurin, Merck, Darmstadt, Germany). All cells (MG63, HBMSC and HDFn) were seeded into 12-well plates at 3 × 10^5^ cells/well. After 24 h, maximum concentration of 50% of particle solution with cell culture medium (0.015 mg mL^−1^) of the four conjugated QD-systems (QD_Chi_LM and HM, QD_Chi-OPS_LM and HM) were added and incubated for 1 h. Afterwards, the supernatant was removed and a solution with rezasurin (0.1 mg/mL, Merck, Darmstadt, Germany) was incubated with the cell culture for 3 h. Then, the supernatant intensity of fluorescence was quantified with a fluorimeter (Synergy Mx, BioTek, Winooski, VT, USA). The cells were kept in culture with only complete medium. As control, cells were cultured with basic medium on the well-plate with the same time-points. To further confirm the growth and viability, a live/dead test was performed with the three different cell types that were cultured until the same time-points. Cells were stained with calcein-AM and propidium iodide (Live/dead assay, Merck, Darmstadt, Germany) to determine the viability of the different cells. Then, the living cells were imaged using a Zeiss Inverted Fluorescence Microscope (IFM, Axio Imager Z1, Jena, Germany). A quantitative biocompatibility evaluation with the MG63 cells was also performed with Annexin V-FITC Apoptosis Detection Kit (Merck, Darmstadt, Germany) as described by the manufacture and evaluated by flow cytometry analysis (BD FACSCanto™ II, BD Bioscience, San Jose, CA, USA).

#### 2.5.2. Flow Cytometry Single-Cell Analysis (FACS)

For FACS, HBMSC, HDFn, and MG63 cells were seeded and treated as described for the Alamar blue assay. To avoid the endocytosis, cells seeded for 24 h were pre-incubated (30 min) at 4 °C and consecutively four conjugated QD-systems (QD_Chi_LM and HM, QD_Chi-OPS_LM and HM) were added and incubated for 1 h (0.015 mg mL^−1^).

Subsequently, cells were detached by trypsin–EDTA solution (0.25%, Gibco) and suspended in PBS buffer for analysis. BD FACSCanto™ II (BD Bioscience, San Jose, CA, USA) where lasers at 405 nm and 488 nm wavelength were used to quantify (2 × 10^4^ cells) the positive cells for QD uptake. The median of the cell fluorescence distribution (X-mean) from each experiment was normalized to X-mean of the untreated control. Same control as described before (TCPS) were analyzed. 

#### 2.5.3. Vesicular Trafficking Evaluation

HBMSC and MG63 cells were seeded at 5 × 10^4^ cells/well (24 well-plate) on tissue culture cover slips (13 mm diameter) for 24 h at 37 °C and 5% CO_2_. The four conjugated QD-systems (QD_Chi_LM and HMW, QD_Chi-OPS_LM and HM) were added to the cells for 1 h at 37 °C in 5% CO_2_ (0.015 mg mL^−1^). Afterwards, cells were stabled with paraformaldehyde (4%) and the early endosomes were targeted with Early Endosome Marker (Anti-EEA1 antibody—ab2900, Abcam, Cambridge, UK) and secondary antibody Alexa-Fluor 594 nm (Life Technologies, Invitrogen, Waltham, MA, USA) solution and covered with fluoromount. Laser confocal scanning microscopy (Leica SP5II, Leica Microsystems Inc., Wetzlar, Germany) evaluated the cells with lasers wavelengths at λ = 488 nm and 594 nm, to stimulate the different nanoprobes. Same control was used as described before (TCPS, Corning, NY, USA). 

#### 2.5.4. Osteogenic Differentiation Evaluation

HBMSC and MG63 cells (5 × 10^3^ cell/well) were incubated with osteoinductive medium (0.1 mM dexamethasone, 0.1 mg·mL^−1^ ascorbic acid and 10 mM b-glycerophosphate) together with different QD labels (QD_Chi_LM and HM, QD_Chi-OPS_LM and HM) and were characterized by histochemical analysis for Alizarin red S (calcium deposits), alkaline phosphatase activity (ALP), collagen type I, and Von Kossa (phosphate deposits) staining. 

ALP was measured as quantitative analysis for early-osteogenic differentiation characterization. After 3 weeks, QDs labeled and unlabeled HBMSC and MG63 cells were detached and lysed for 30 min (Triton X-100, 1%, Merck, Darmstadt, Germany). The enzyme activity was assayed by p-nitrophenol phosphate (Merck, Darmstadt, Germany) hydrolysis at pH 10.5. The solution color was stabilized by NaOH (0.02 M; Merck, Darmstadt, Germany), and p-nitrophenol was quantified at 405 nm by a plate reader (BioTek, Winooski, VT, USA). The ALP activity results were standardized to total protein content (Lowry’s method) and were expressed in nanomol/min/µg of protein.

#### 2.5.5. Osteogenic Differentiation-Related Gene Expresion (RT-PCR)

Total RNA was extracted from lysed cells supernatant using the NucleoSpin kit (Macherey-Nagel, Dueren, Germany) according to the manufacturer’s guidelines. RT-PCR reaction and amplification (Titan One Tube RT-PCR system; Roche, Branchburg, NJ, USA) were followed by specific temperature for denaturation (94 °C), annealing (55 °C), and elongation (68 °C). Table 1 shows the primer sequences for RT-PCR protocol. Results were expressed as normalized ratios by values for housekeeping gene (glyceraldehyde 3-phosphate dehydrogenase—GAPDH). The RT-PCR yields were divided by electrophoresis agarose gel (1%) and stained by GelRed™ Nucleic Acid (fluorescence—Biotium, Fremont, CA, USA) and analyzed with a cross-platform image analysis software (Fiji).

Quantitative real-time PCR was carried out in mixture containing 1 μL of cDNA, 10 μM of each forward and reverse primers for osteocalcin, bone morphogenetic protein 2 and osteopontin (Table 1) and 10 μL of iTaqTM Universal SYBR^®^ Green Supermix (BioRad, San Jose, CA, USA). qPCR experiments were run, using an iQ5 (BioRad, San Jose, CA, USA) and analyzed with the iCycler IQ software (BioRad, San Jose, CA, USA). The housekeeping gene (GAPDH) was used as the endogenous assay control. Relative quantification of gene amplification by qPCR was performed using the cycle threshold (Ct) values and relative expression levels were calculated using the 2^−ΔΔCT^ method. For each PCR, samples were analyzed in duplicate and three independent experiments were performed.

#### 2.5.6. Different mRNA Expression of Caveolin-1 (Real-Time PCR)

qRT-PCR experiments were performed as describe above (Section 2.5.5) and samples were processed at iQ5 (BioRad, San Jose, CA, USA) and analyzed with the iCycler IQ software. GAPDH was used as the endogenous assay control. Relative quantification of gene amplification was performed as described before and the expression value for the target gene Caveolin-1 (Table 1) was normalized to the GAPDH value at each time point. Results were normalized to the HDFn average results and are represented as fold change. 

#### 2.5.7. Hemolytic Activity

Human residual blood cells from fresh blood of healthy individuals were purchased (Centro Hospitalar de São João, Porto, Portugal) and processed to obtain hemocytes with a density gradient media. The purified hemocytes (6 × 10^8^ cells/mL) were distributed in a 96-well plate. Chi and Chi-OPS (LM and HM) QD’s sequence dilutions were added at a range of 5 to 50% (0.0015–0.015 mg mL^−1^) After 1 h incubation, hemocytes’ supernatant was collected and the released hemoglobin was determined at 450 nm (Synergy Mx, Biotek, Winooski, VT, USA). Untreated hemocytes were used as negative control and positive controls were incubated with 0.2% Triton X-100. The hemolysis percentage was calculated as
((sample absorbance − negative control absorbance)/(positive control absorbance − negative control absorbance)) × 100.

### 2.6. Statistical Analysis

One-way ANOVA test (GraphPad software, La Jolla, CA, USA) was used and differences between samples were considered statistically significant when *p* < 0.05. 

## 3. Results

### 3.1. In Vitro Evaluation of CdS/Polysaccharide Conjugates 

As a target probe, potential cytotoxicity of QD-conjugated nanoparticles stimulated extensive research, because the heavy metal core (CdS) and the QDs’ biological safety relies on its effect on cellular function and metabolic activity. As a result, the main goal of this study was to develop compositions with different capping ligands with “green chemistry” for CdS QDs compatible potential use for cell/tissue applications. Chi was conjugated with the OPS to promote specific bio-affinity (QD_Chi-OPS).

In the UV–Vis spectra of CdS nanoparticles (Figure 1A), absorption onset of the S0→S1 (λ_exc_) were detected in the range of 375–390 nm, “blue-shifted” to CdS bulk value (λ = 512 nm or 2.42 eV) [24], indicating the formation of nanoparticles in quantum confinement regime. The band gap energies (EQD) estimated using the linear region of TAUC plot [25] (inset in Figure 1A) were 2.80 ± 0.05 eV for all synthesized CdS QDs, independently of the capping ligand (within statistical variation). As expected, nanoparticles increased the band gap of QDs and created low energy bands, these band gap results are higher than the CdS reference bulk value of 2.42 eV [24]. The mean sizes of CdS nanoparticle (2R = 2.3 ± 0.2 nm) was determined using Henglein’s empirical equation [26], which correlates the diameter (2R) of the core of nanoparticle to the excitation intraband absorption (exc). 

The QDs’ PL spectra were measured at room temperature and stabilized. Chi and Chi-OPS (LM and HM) spectra are presented in Figure 1B. For all systems, band-to-band optical transitions were not detected and QDs showed different colored light emissions (blue to red) coming mainly from defect activated photoluminescence.

TEM analysis (Figure 2A,B for QD_Chi_LM and QD_Chi-OPS_LM, respectively) indicated the presence of crystalline QDs with spherical shape and monodisperse distributions, the continuous lattice fringes resulted by the electron diffraction pattern indicated CdS diameters ranging from 2.5 nm to 3.5 nm.

It is widely known that, besides the chemical composition, charge, size, and surface chemistry of nanoparticles have important roles in the biological responses. Thus, the surface charges of the synthesized nanoconjugates were determined by electrochemical equilibrium measurements and the values ranged between +30 mV and +38 mV (Figure 2C). These results indicated the predominance of positively charged surfaces due to the protonated amino groups (R- NH_3_^+^) of cationic chitosan. Furthermore, the results showed the relative reduction of the average positive ZP values after conjugation with OPS, a slight influence of molecular mass and DD of chitosan polymer used. In addition, the ZP results showed higher than positive 30 mV by the QDs (CdS core) that were stabilized with electrostatic forces by Chi and Chi-OPS as colloidal nanoconjugates (shell). Moreover, DLS technique evaluated the hydrodynamic diameters (HD) of the surface charged QDs in the aqueous medium (Figure 2D). After the synthesis, the amount of contributions of QD inorganic core with the outer ligand (Chi or Chi-OPS) and its medium interactions resulted in a perfect sphere diameter ranging from 21 nm (QD_Chi_LM) to 30 nm (QD_Chi_HM), which visibly showed a relative higher Chi_HM solvation volume. For Chi_LM the conjugation with OPS increased HD of the nanoconjugates but for HMW polymer the results were similar before and after conjugation. Nonetheless, the HD values of the nanoconjugates remained relatively small (<30 nm) for all the systems. 

Based on these results, the QDs synthesized with chitosan of low and high molecular weight presented similar physicochemical and morphological features (Figure 2E,F). Thus, it is expected that the biochemistry of the QD-chitosan functionalized by OPS will determine the influence on the biological behavior of the nanoconjugates.

### 3.2. Fluorescent Bio-Nanoprobes Biocompatibility Evaluation

In this work, the CdS-based quantum dots were created to target a specific cell phenotype to be followed over time. For that application, the OPS was chosen as being part of some non-collagenous proteins of bone tissue (such as osteopontin and osteonectin) and their specific bio-affinity for bone-derived cells. OPS was joined with the Chi (Chi-OPS) and produced as surface coating ligand of CdS QD (QD_Chi-OPS). The specificity for Chi-OPS application was compared to the system using polymer, without OPS as surface ligand (QD_Chi). Human bone marrow cells (HBMSCs) were isolated by the adherent method on plastic surface. In order to confirm whether prepared HBMSCs were able to differentiate into osteogenic lineage, we observed the nanoparticles effect in the cells’ differentiation into osteoblast phenotype after biological osteoinduction. Along the different periods of time (7, 14, and 21 days), cellular toxicity was not observed in all human cell types such as HBMSCSs, human osteoblast-like cells (MG63), and human neonatal dermal fibroblasts (HDFn), at concentration of 50% of CdS_QD nanoparticles (0.015 mg/mL), conjugated either with QD_Chi or QD_Chi–OPS with low and high molecular weight (LM and HM) (Figure 3).

Our findings strongly support the cytocompatibility of the different bioconjugated QDs (approximately 100% cell viability) with and without OPS for both HBMSC and MG63, after 14 and 21 days (Figure 3). Concerning the biological characterization, our results of cytotoxicity strongly support the biocompatibility of the different bioconjugated QDs (approximately 100% cell viability) with and without OPS for all cell types. Live/Dead bioassay with calcein and propidium iodide incubation corroborated the cellular viability using the Alamar blue assay (Figure 4). Similar results of cellular cytotoxicity of the lower molecular weight bioconjugate particles were shown in a previous work [23]. These results could suggest that the OPS-modified nanoparticles could be reliable and safe to be used in vitro and perhaps in vivo in pre-clinical analysis, depending on the biomedical application.

In this study, the qualitative calcein (live cell) assay was used as a corresponding test to support the biocompatibility results of cellular metabolic activity by Alamar blue (Figure 3). To reduce the volume of flow cytometry analysis, the apoptotic detection test was performed only on MG63 cells. The quantitative cytotoxicity method provided by a cell apoptotic assay of two-color fluorescence, that quantifies the early apoptotic cells (Annexin V—stained in green—FTIC filter) and dead cells (propidium iodide—red cells—PE filter) by FACS. All CdS_QD conjugates results showed similar results to the control group (TCPS) (Figure 5). Low apoptotic cell presence corroborates to the cell viability assays (rezasurin) and calcein stain of CdS nanoconjugates with MG63 cells, which was not influenced by the OPS modification or chitosan molecular weight (LM or HM, Figure 5).

### 3.3. Cellular Uptake of QD Nanoconjugates 

To evaluate the specificity of QD nanoprobes, neonatal human dermal fibroblasts (HDFn) that are mesenchymal cells from the mesoderm embryonic-derived precursor cells and secrete type I and/or type III collagen that will form a soft ECM [27]. HBMSC were chosen as a primary cell line, isolated from bone marrow and MG63 that are human osteoblast-like cell line derived from an osteosarcoma were chosen to ensure high cell viability during cell culture and turn over similar to the other human bone-derived cells [26]. The cellular behavior regarding the nanoparticles’ uptake, intracellular mechanisms and endosomal escape, was followed by confocal laser scanning microscopy (CLSM) and FACS. For these studies, cells were cultured with QD-conjugated particles for 1 h (0.015 mg mL^−1^), afterwards, cellular nanoparticle internalization were qualitatively and quantitatively analyzed. Images in Figure 6 and Figure 7 show a high number of HBMSC cells QD_Chi-OPS positives (LM and HM ~30%) and also the osteoblast-like cells (MG63) were positive for the QD_Chi-OPS presence (LM ~40% and HM ~50%). However, lower percentages of stromal cells were positive for QD_Chi nanoprobes (HBMSC < 15%). 

The QD_Chi-OPS nanoprobes uptake by the fibroblasts (HDFn) were not different when compared to the bone cells (Figure 6 and Figure 7). Thus, the HDFn different particle uptake results for the probes were all low and very similar (>15%—Figure 6 and Figure 7), regarding to a non-specific particle uptake, without influence of the OPS-modification.

In the endocytosis evaluation, the chitosan different molecular weight did not change significantly the cellular endocytosis by the cells, but the surface modification (OPS) had an important role in HBMSC particle uptake. Some enhancement in the MG63 cells uptake of Chi_ and Chi-OPS_QD could be observed with the CdS Chi_HM (Figure 6 and Figure 7). We could infer that the OPS surface modification of the particles caused higher uptake with statistical difference between the same chitosan molecular weight in the same cell type. It seems that the presence of OPS improved cellular uptake to both bone-derived cells tested. However, the endocytosis pathway and particle intracellular trafficking were inhibited when the percentage of positive cells (HBMSC and MG63) decreased to values below 5% of QD-positive cells with their incubation at lower temperature (4 °C—Figure 7).

The two major purposes of this work were to promote intracellular accumulation of QDs in the bone marrow stromal cells and to enhance particles endosomal escape, to keep a strong fluorescence signal of each cell. QD localization within the cell cytoplasm was analyzed by Confocal Laser Scanning Microscopy (CLSM), after 1 h of CdS_QDs bioconjugated solution incubation with HBMSC cells. As shown in Figure 8, for that time point, QD particles were not accumulated at the cell’s surface (HBMCS), while a higher amount of the nanoprobes had been stored within the cytoplasm, the nanoparticles (green) had an intracellular position coincident with the early endosomes (stained in red), but the image does not have high enough definition, due to the intense background of CdS_QD’s on the different laser channels. In accordance with the above-mentioned flow cytometry results, only the QD_Chi-OPS particles could be found within a high number of cells, dispersed all over the cell cytoplasm (green—Figure 8). The QD_Chi-OPS suggested that their intense intracellular presence could indicate that there is a specific pathway for the exhaustive cell uptake. The HBMSC intracellular trafficking was associated to cells endocytosis (Figure 7) that could be driven by their movement through the endosomal complex (Figure 8). By contrast, the HBMSCs that uptake Chi-LM and Chi_HM nanoparticles did not show a similar response, as the fluorescent particles were dispersed at the bottom of the well plate, but it was not possible to find a single cell with the co-location of QD particles inside the early endosomes within cell cytoplasm (Figure 8).

In the Caveolin 1 expression RT-PCR results with HBMSC, MG63, and HDFn, the difference between the different cell lines cultured with the Quantum dots is not completely evident (Figure 9), the gel bands are very similar between HBMSC’s and MG63 cells. However, with the real-time PCR analysis, the mesenchymal stromal cells (HBMSC) showed higher expression of Caveolin 1, which corroborates the higher Chi-OPS QD’s particles uptake by these cells (Figure 6 and Figure 7). 

### 3.4. HBMSC’s Osteogenic Differentiation Evaluation

In the osteogenic differentiation assays, after the pre-incubation time (particle uptake), Chi and Chi-OPS CdS QDs-labeled and unlabeled HBMSCs and MG63 were cultured in 2D cell culture plates for 21 days. Both cell types labeled or not with different QD’s showed high ALP activity and they were successfully differentiated into early-differentiated osteoblasts (Figure 10). 

Histochemical assays were performed according to the osteogenic differentiation protocol. Calcium deposits promoted by osteoblasts were evidenced by Alizarin red staining (Figure 11), Von Kossa protocol was also used to stain the phosphate deposits to evidence the ECM mineralization (late marker for osteogenic differentiation). ALP activity and Collagen type I were observed as early markers of osteogenic differentiation. Similar histochemical images of osteogenic differentiation of HBMSCs with and without different CdS_QDs particles were observed (Figure 11), but in the labeled HBMSC’s with OPS-modified nanoparticles, an enhancement on the calcium deposits was observed, when compared to the other nanoparticles (without OPS) and control (TCPS). The high molecular weight Chi-OPS nanoparticle presented the highest mineralized ECM (Alizarin red and Von Kossa staining).

The gene expression of HBMSC and MG63 showed cellular differentiation after 21 days in culture with high expression of ALP, BMP-2, and Runt-related transcription factor 2 (Runx-2). Thus, there was no difference between the osteogenic gene expression between the cells incubated with the different nanoparticles (Chi_ and Chi-OPS_QDs) (Figure 12A,B). All the osteogenic genes tested in different stages of differentiation (Runx-2, BMP-2, osteoprotegerin (OPG)) were similarly expressed by the tested cells with and without the OPS-modification or with different Chi molecular weight. These results showed that the particle uptake did not directly influence the osteogenic differentiation of HBMSC and MG63. Late-osteogenic differentiation by the gene expression of MG63 and HBMSC after 21 days (osteocalcin, osteopontin and BMP-2) was also evaluated. The results shown in Figure 12C,D presented an enhancement on the expression of osteocalcin by the HBMSC that were incubated with the OPS-modified nanoparticles (Figure 12C). Similar behavior was not observed by the MG63, that showed similar osteocalcin expression for all the tested nanoparticles (Figure 12D).

### 3.5. Hemocompatibility Assay 

The four bioconjugated nanoparticles caused no significant hemolysis, no change in size or count of red blood cells within the studied dose range when compared to the positive control tested (Triton X-100 at 0.2%—Figure 13). The results obtained in this work are fairly probable as synthetic materials usually do not affect the integrity of red cells, which are very sensitive to amphiphiles or polyanionic surface and polycations [27].

## 4. Discussion

As a target probe, QD-conjugated particles cellular biocompatibility is an important aspect to research, because the internal metal within QDs (CdS) and the biological application depend on its effect on cell function and viability. Consequently, the main objective of this study was to design and produce tunable and biocompatible CdS QD nanoprobes, using chitosan (polysaccharide with different molecular weight) as direct capping ligand by green chemistry (water route at pH 7 and at RT) to be used as biomedical tools. Chitosan was conjugated with OPS for a specific bio-affinity to functionalize the CdS QD (QD_Chi-OPS). In the biological characterization, the results of cytotoxicity strongly support the biocompatibility of the different bioconjugated QDs (approximately 100% cell viability) with and without OPS for both cell lines. Similar results were observed in a previous work with the Chi (lower molecular weight, modified with OPS or not) [23,28]. Live/Dead bioassay with calcein and propidium iodine incubation corroborated the cellular viability results for all the different cell types (Figure 3) showing intense cell density remaining viable after 21 days. 

In the endocytosis evaluation, the different molecular weight of the chitosan modification (with or without OPS) did not change significantly the intracellular trafficking of the Chi-CdS_QD by the HBMSC cells. However, some enhancement in the MG63 cells uptake of Chi_ and Chi-OPS_QD could be observed for the higher molecular weight chitosan (Figure 6). This result was unexpected, as we should have observed higher endocytosis with smaller particles [29]. Finally, the endocytosis pathway for cellular particles uptake could be observed when the percentage of positive cells (HBMSC and MG63) decreased drastically in the incubation at lower temperature (4 °C—Figure 7 and Figure 8), as it is already known that temperature is important for membrane particles endocytosis [30]. In fact, especially flow cytometry and confocal imaging results clearly demonstrated the intracellular trafficking of QD_Chi-OPS LM and HM by the HBMSC. The results could also be related to some nanoparticles characteristics, such as size, shape, surface chemistry and charge, that could enable higher or lower cellular uptake depending on the endocytosis mechanism [30], but the mechanisms of endosomal release related to these Quantum dot conjugates are still unclear. The elucidation of these mechanism by which nanoprobes are internalized into the cells could provide insights about the intracellular trafficking, fate and cytotoxic profile of the nanomaterials [30]. Yet, a number of different endocytic machinery can be driven by mammal cells to uptake a large variety of particles. These pathways for non-phagocytic cells include cadherin-mediated or non-mediated endocytosis, the latter including internal trafficking via Caveolae [29]. It is believed that such different types of internalization mechanisms were used by cells to accomplish different tasks. One hypothesis for the MSC’s higher particles uptake could be related to the composition of their membranes, the different organization can affect cell responses to external stimuli and signaling molecules. In fact, membrane lipid rafts are recognized as important platforms regulating activity at the cell surface [31]. Caveolin-1 is a scaffolding protein of cholesterol-rich Caveolae lipid rafts in the membrane. Furthermore, Caveolin-1 has the ability to bind to many cell signaling molecules and regulate cell signal transduction in Caveolae. These are invaginated, flask-shaped plasma membrane domains, which are especially enriched in cholesterol and sphingolipids. They are characterized by the presence of the integral membrane protein Caveolin. Interestingly, Caveolae can internalize large molecular complexes [29]. However, it was shown that the expression of Caveolin-1 increases in MSCs induced by osteogenic differentiation [32]. Caveolin-1 protein is enriched in density gradient-fractionated in the MSC plasma membrane, consisting of ~100 nm diameter membrane-bound vesicles, and is distributed in a punctate pattern. Another interesting fact is that the expression of genes related to Caveolae mediated endocytosis is upregulated in human bone marrow-derived MSCs (HBMSC) compared to hematopoietic stem cells (HSCs), suggesting a functional role for this pathway in MSCs. Similar to MSCs, caveolae are particularly abundant in adipocytes, endothelial cells, and muscle cells, and they are relatively few or completely absent in many human cancer cells [33,34,35]. HBMSC showed higher expression of Caveolin-1, which is in accordance with the literature [32]. These findings suggest that Caveolin-1 normally acts to regulate the differentiation and renewal of MSCs, and increased its expression during MSC ontogenesis, which could be related to higher uptake of Chi-OPS QD’s particles. Thus, the presence of OPS on the surface of the Quantum dots should also favor the nanoprobes interaction with the HBMSC membrane (Caveolae), thus enhancing the intracellular particle trafficking by MG63 and HBMSC.

The MSCs ability to differentiate into specific lineages is of common knowledge, and their capacity to establish a bone cell phenotype is one of most studied topics in bone tissue repair [36]. Encouraging results have been shown when using osteoblasts derived from MSC osteogenic differentiation in bone regeneration [37,38]. In this work, we also tested the possible effect of Chi and Chi-OPS CdS QDs on the HBMSC and MG63 cells differentiation capacity since the genotoxicity of this nanoprobes was not completely explored. Therefore, the capacity of HBMSCs to differentiate into bone tissue was explored by analyzing many osteogenic-specific markers. First, the measurement of ALP expression is a model for early marker that enables to follow the stage of osteoblastic differentiation. This enzyme activity is increased during bone ECM synthesis, which corresponds to the early-stage of cell osteogenic differentiation [39]. In all cells tested with and without QD’s, the ALP activity was very similar, showing small increase in the cells incubated with OPS-modified nanoparticles.

Several specific transcription factors are key readers of the multipotent mesenchymal cell gene expression activation into the osteoblast phenotype. Runx-2 is an encoding gene of several bone matrix proteins, such as osteopontin and osteocalcin, leading to a higher pre-osteoblasts presence from stem cells differentiation [40]. The progressive development of the osteoblastic phenotype (osteogenic-marker) from an undifferentiated cell to completely differentiated osteoblast is characterized by specific genes expression that presented periods of osteoblast phenotype differentiation: late stage of osteoblastic differentiation, ECM synthesis and mineralization [41]. OPG is synthesized by mature osteoblasts. The most significant effect driven by OPG is to reduce osteoclast maturation and activity, to promote bone formation [42]. All the osteogenic genes tested in different stages of differentiation (Runx-2, BMP-2, OPG) were similarly expressed in all cells tested with and without the presence of different bio-nanoprobes. However, regarding the quantitative PCR results, the HBMSC showed that the OPS-modified particle uptake enhanced the osteogenic gene expression by those cells (osteocalcin). Regarding the OPS signaling, previous work where this molecule was immobilized on collagen/nanohydroxyapatite scaffolds showed the induction of MSCs from different tissue sources to differentiate into an osteoblast phenotype [20,21]. 

Blood and lymphatic vessels of the body circulatory system are vital for organ and tissue function, but are present in many pathological processes, including solid and metastatic cancer. Most of the nanoprobes are injected intravenously, so the blood biocompatibility is an important characteristic to this kind of nanoparticles. We evaluate the different CdS QDs for blood cellular response. Results showed hemotolerance by all tested nanoparticles with respect to most of blood’s cellular components. Hemolysis assay (or red blood cell lysis) is the most popular parameter adopted by some authors [43]. A recent study [27] showed that Cd-containing QDs with surface biofunctionalization of Chi as a biocompatible shell-capping ligand presented in vitro cytotoxicity responses, remarkably depending on the cell type, concentration, and period of exposure to the colloidal nanoconjugates. The concentration of CdS nanoconjugate was the predominant factor determining its toxicity followed by the time of incubation tested with different cell types [28]. However, a meta-analysis of cellular toxicity for cadmium containing quantum dots showed that toxicity is closely correlated with quantum dot surface properties (including shell, ligand and surface modifications- protein, amino acid, peptide or polymer), diameter, assay type and exposure time to the cells [44]. The fluorescent CdS nanoconjugates with chitosan-OPS shell were made using a “Green chemistry” and showed appropriate cell biocompatibility, with Caveloae interaction to track bone marrow MSCs, and pertinent properties for bioimaging into a pre-clinical animal model. The use of this nanoprobe could benefit some applications in studies of regenerative medicine and oncology such as bone cell tracking and metabolism elucidation of nanoparticles uptake, mechanism of bone cell differentiation triggered by smart biomaterials; target tumor cells from bone cancer or metastasis in bone tissue.

## 5. Conclusions

In this work, it was shown that nanoparticles produced with CdS QDs as inorganic core with surface modification by chitosan covalently linked O-phospho-L-serine as a bioactivity shell-capping ligand, is non-cytotoxic in vitro, within the studied concentrations and times of culture. Moreover, the cellular metabolic activity results were not affected by the cell tissue source, where HBMSC, MG63, and HDFn cells were viable and proliferated after incubation with the different CdS QD bioconjugates. The chemical biofunctionalization of QDs with Chi-OPS favored higher uptake of bone tissue-derived cells such as primary HBMSCs and osteoblastic-like cell line (MG63). Regarding flow cytometry results, after QD_Chi and _Chi-OPS bioconjugates had been internalized by the cells (HBMSC and MG63) via endocytosis, the QDs were uniformly distributed throughout the cellular content. Microscopy images showed that the highest numbers of labeled cells (HBMSC) were observed for the incubation of QD_Chi-OPS solution (LM and HM), giving robust indication that the OPS-modification favored the particles intracellular trafficking by HBMSC. But, concerning the osteogenic differentiation, the CdS QD Chi-OPS_HW promoted higher osteocalcin gene expression and ECM mineralization, dealing with a late stage of differentiation for the osteoblast phenotype. On the contrary, dermal fibroblasts (HDFn) showed lower particle uptake, not influenced by the OPS modification and different molecular weight. Moreover, this intramembranous delivery should contribute to enhance the cell fluorescence intensity and could favor bioimaging application. The use of Chi-OPS_QDs nanoprobes associated with advanced imaging techniques indicated that they could be used to identify exogenous cells interactions during bone tissue engineering transplant processes. Moreover, in a future study, the fluorescent cells could allow longitudinal monitoring of cell recruitment, proliferation into bone environment by the specific-target of QD nanoprobes to validate a paradigm that can be translated to investigate other regenerative therapies in pre-clinical models.

## Figures and Tables

**Figure 1 materials-14-04422-f001:**
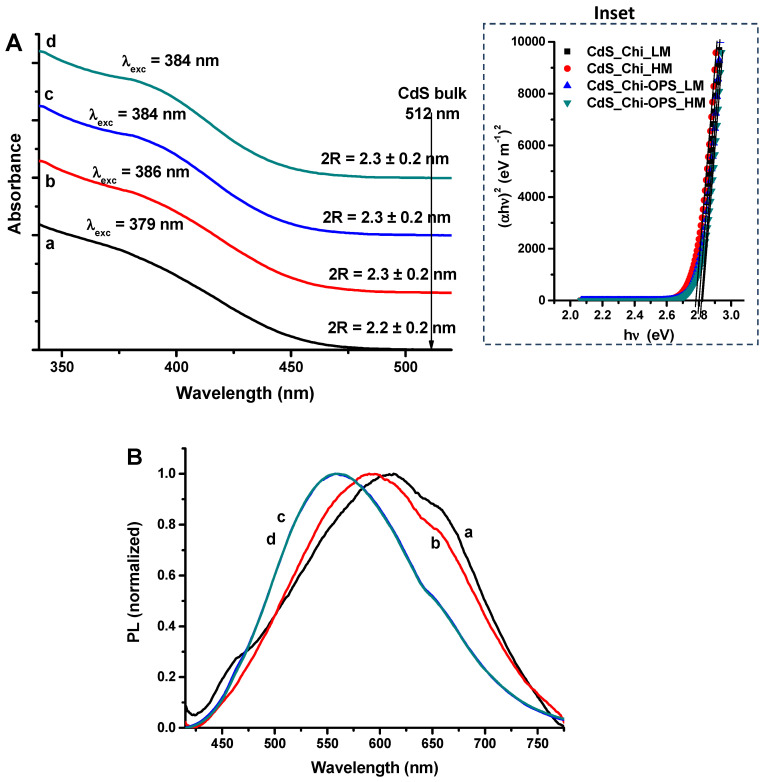
(**A**) UV–Vis and (**B**) PL spectra of (a) QD_Chi_LM, (b) QD_Chi_HM, (c) QD_Chi-OPS_LM, and (d) QD_Chi-OPS_HM. Inset: TAUC relation for CdS QDs. ((c) and (d) spectra are overlapped in PL graph).

**Figure 2 materials-14-04422-f002:**
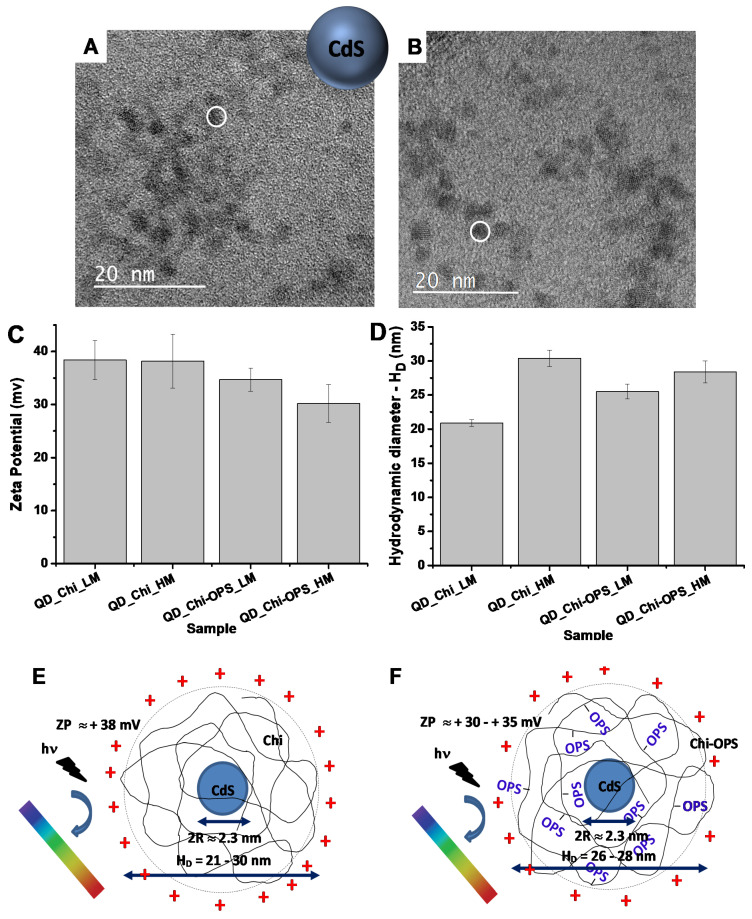
TEM images of QD CdS core (surround by white circle): (**A**) Sample QD_Chi_LM and (**B**) sample QD_Chi-OPS_LM. (**C**) Zeta potential and (**D**) hydrodynamic radius values for Chi and Chi-OPS CdS (LM and HM). Schematic representation of (**E**) QD_Chi and (**F**) QD_Chi-OPS (not to scale).

**Figure 3 materials-14-04422-f003:**
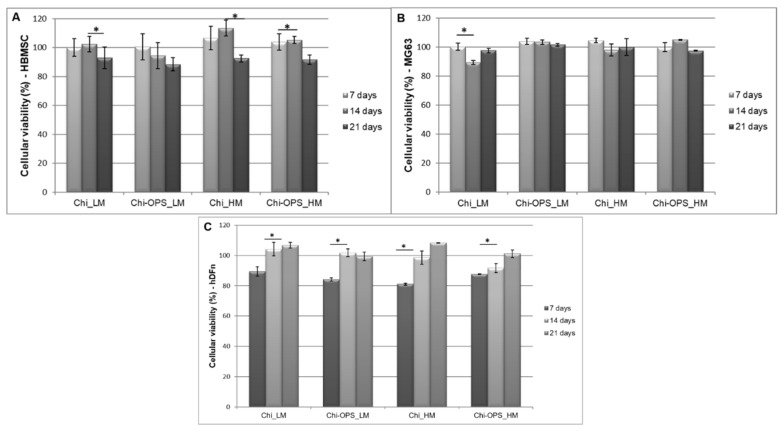
Fluorescent responses of primary cells cultured (HBMSC—**A**), and both cell line cultured (MG63—**B** and HDFn—**C**) incubated with CdS QD_Chi and QD_Chi–OPS (LM and HM) nanoparticles associated with the cellular metabolic. Cell culture control (TCPS) = 100%. Statistical analysis by one-way ANOVA, * *p* < 0.05.

**Figure 4 materials-14-04422-f004:**
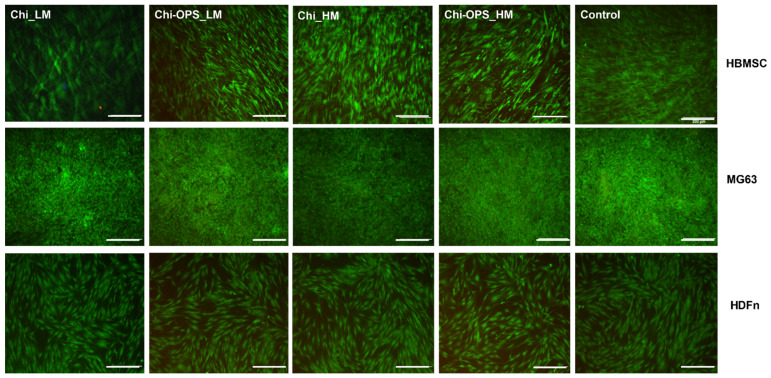
Live/Dead analysis for cell viability (HBMSC, HDFn and MG63) cultured for 21 days. Calcein and Propidium iodide staining in green and red, respectively. Scale bar: 200 µm.

**Figure 5 materials-14-04422-f005:**
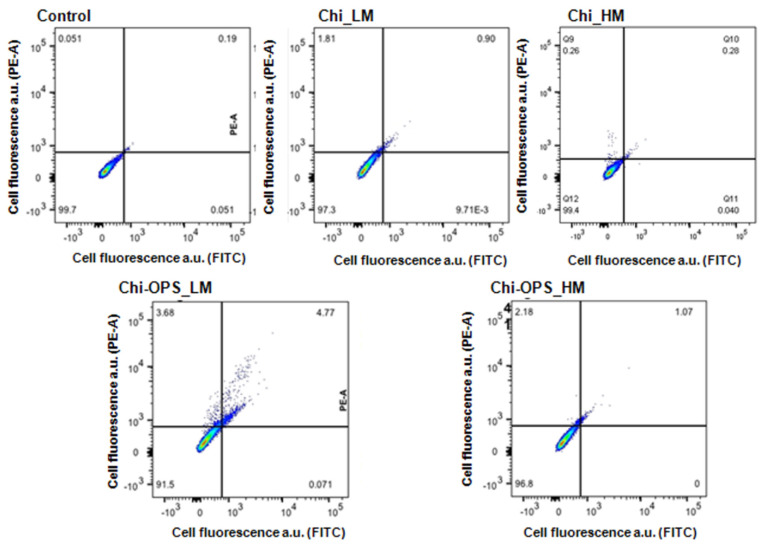
Quantitative flow cytometry study for live/dead assay using different QD conjugated nanoparticles on MG63 cells. FTIC for early apoptotic cells and PE-A for late apoptotic and dead cells.

**Figure 6 materials-14-04422-f006:**
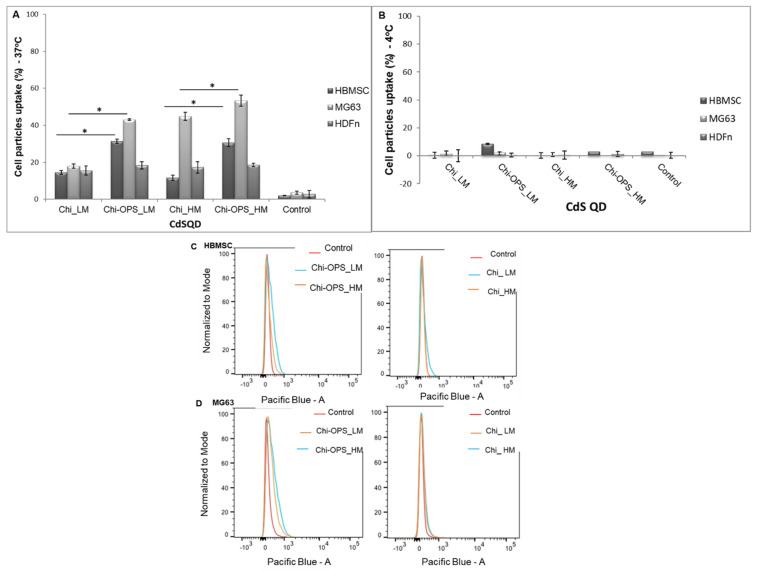
FACS evaluation of QD conjugated nanoparticles on HBMSC, MG63 and HDFn) cell uptake response. The results were normalized based on the untreated control (X-mean of TCPS). (**A**) Cell culture at 37 °C for the incubation period of time, to allow cellular uptake. (**B**) Cell culture at 4 °C for the incubation period of time, to block cellular uptake. Histograms of quantitative flow cytometry results with HBMSC (**C**) and MG63 cells (**D**). Statistical analysis, * *p* < 0.05.

**Figure 7 materials-14-04422-f007:**
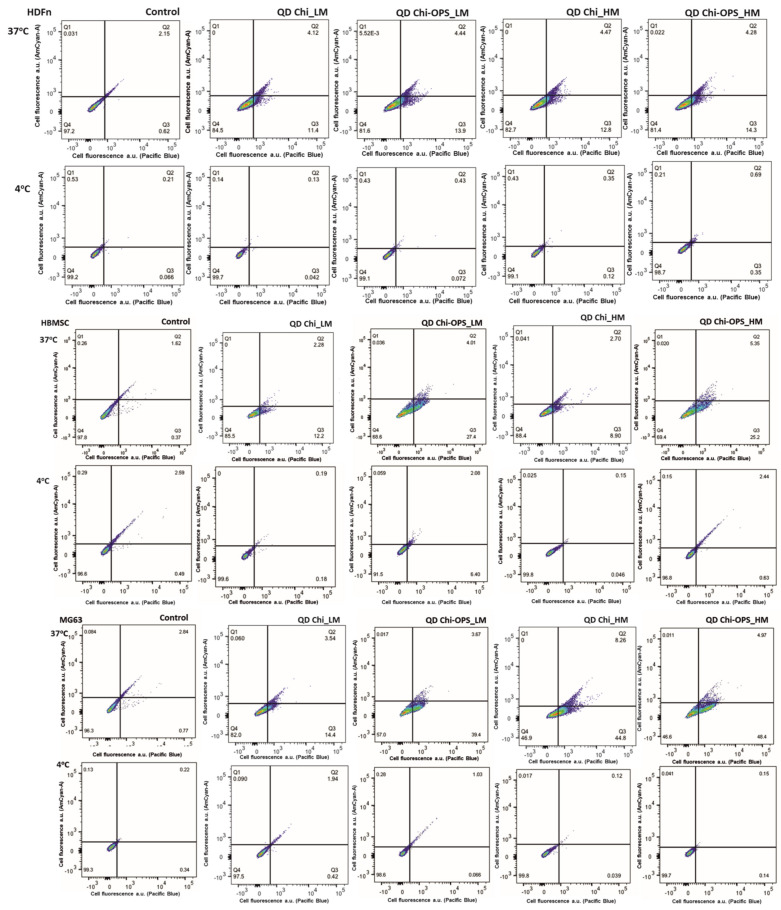
Quantitative flow cytometry study of endocytosis pathway (thermal effect) with different QD conjugated nanoparticles on human dermal fibroblasts (HDFn), human bone mesenchymal stromal cells (HBMSC) and human osteoblast-like cells (MG63) uptake response.

**Figure 8 materials-14-04422-f008:**
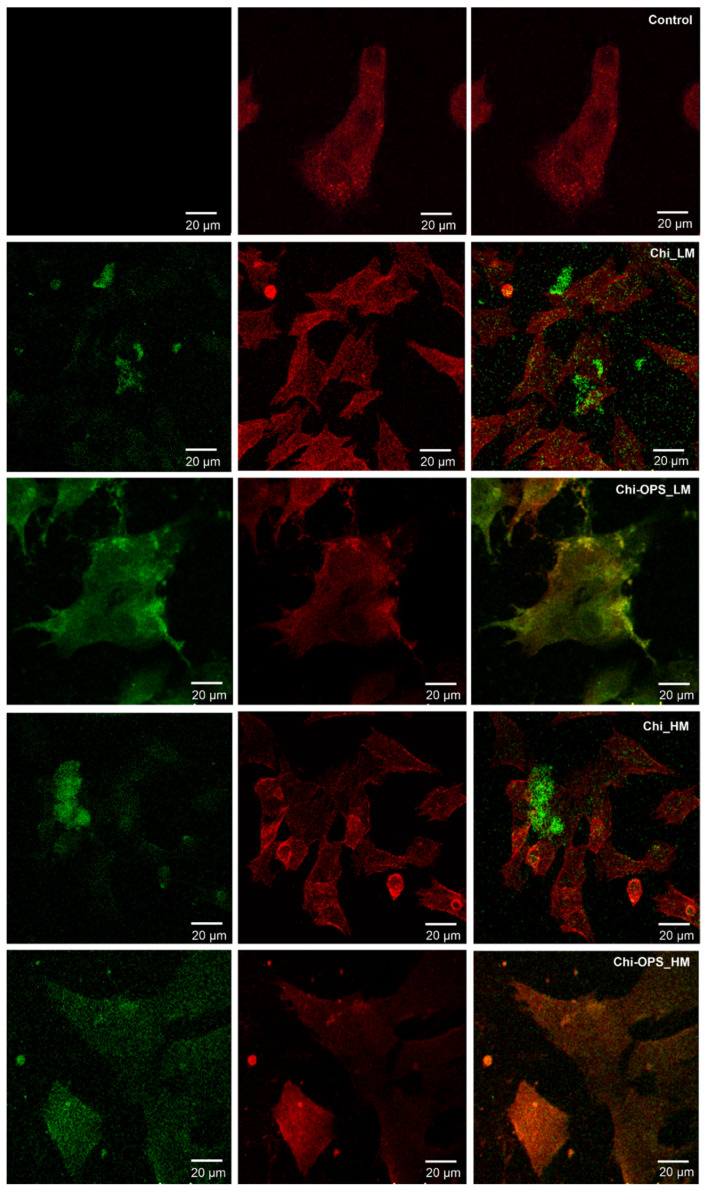
Fluorescence Microscope images of the HBMSC cells nanoparticles uptake. CdSQD particles (green) and early endosome (red) were stained. Scale bar: 20 μm.

**Figure 9 materials-14-04422-f009:**
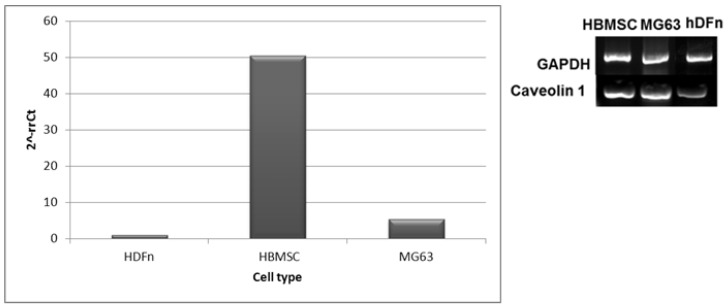
Reverse transcriptase PCR results with human stromal bone marrow cells (HBMSC), human osteoblast-like cells (MG63), and dermal fibroblasts (HDFn) (right image). qPCR of Caveolin-1 using the ∆∆Ct method using GAPDH house-keeping gene expression. Results were normalized to the cells controls (basic medium) and are shown as fold change (left graph).

**Figure 10 materials-14-04422-f010:**
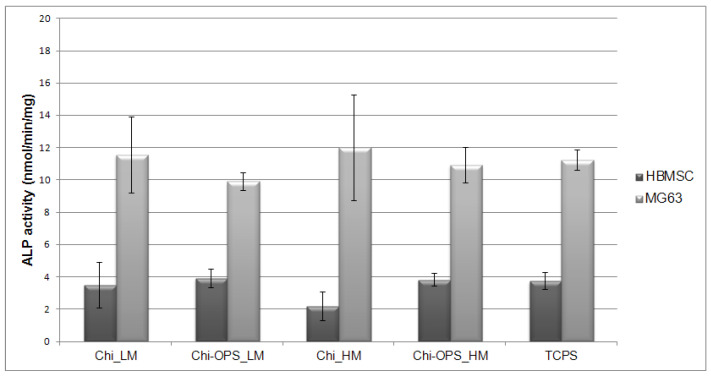
ALP activity of (HBMSC and MG63) cells cultured in osteoinductive medium for 21 days. Statistical differences between samples from same cell source.

**Figure 11 materials-14-04422-f011:**
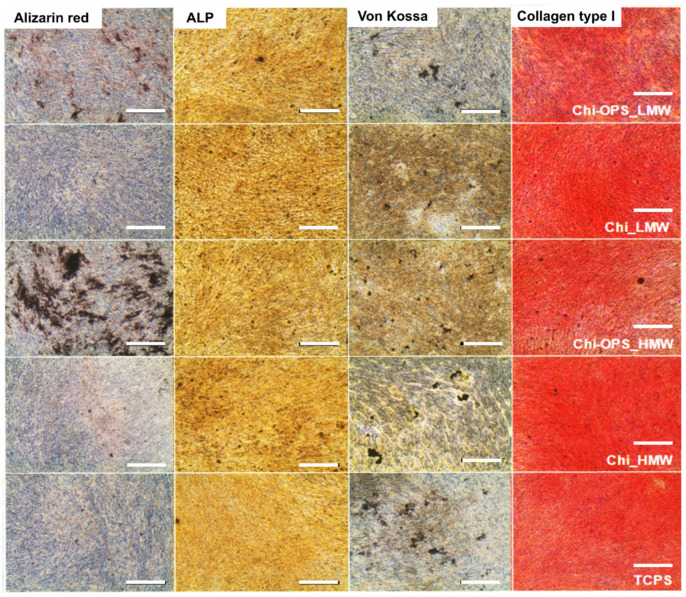
Histomorphometric analysis for osteogenic differentiation of HBMSC cells cultured in osteoinductive medium for 21 days. Scale bar: 200 µm.

**Figure 12 materials-14-04422-f012:**
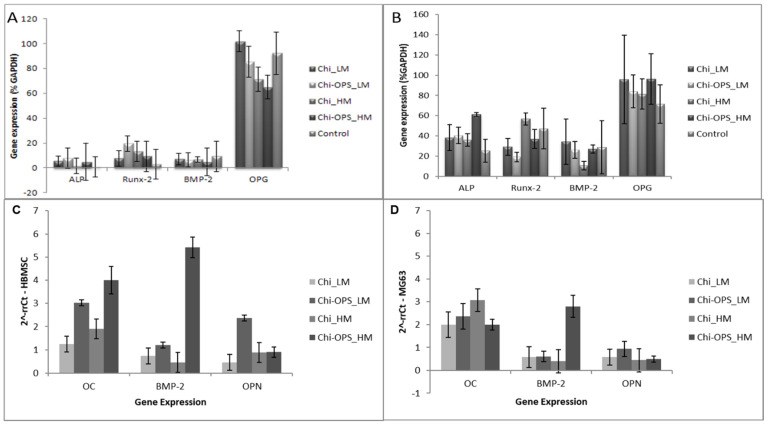
RT-PCR results after 21 days of HBMSCs (**A**) and MG63 (**B**) cells culture under the osteoinductive condition. (**C**,**D**) qPCR of the HBMSC osteogenic gene expression (osteocalcin—OC; Bone morphogenetic protein 2—BMP-2; Osteopontin—OPN) after 21 days. The 2^−rrCt^ method was used with the expression of the GAPDH gene as an endogenous reference. cDNA from HBMSC (C) and MG63 (D) cells grown in TCPS (P4) were evaluated as a negative control (Ref-1).

**Figure 13 materials-14-04422-f013:**
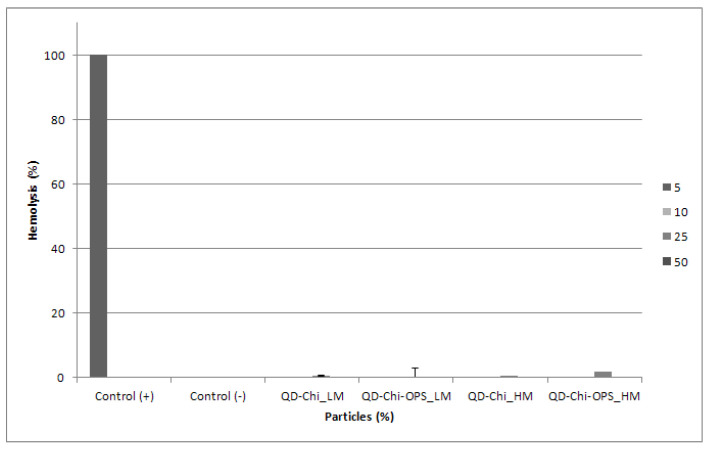
Hemolytic activity of different QD conjugated nanoparticles on human red blood cells. PBS—negative control and 0.2% Triton X-100—positive control.

**Table 1 materials-14-04422-t001:** Primers for PCR Amplification.

Gene	Forward Primer	Reverse Primer
GAPDH	5’-TAACTGGTAAAGTGGATATTG-3’	5’-GAAGATGGTAGATGGATTTC-3’
Runx-2	5’-GTGCCTAGGCGCATTTCA-3’	5’-GCTCTTCTTACTGAGATGGAAGG-3’
OC	5’-AGAGTCCAGCAAAGGTGCAG-3’	5’-TCAGCCAACTCGTCACAGTC-3’
ALP	5’-ACGTGGCTAAGAATGTCATC-3’	5’-CTGGTAGGCGATGTCCTTA-3’
BMP-2	5’-GACGAGGTCCTGAGCGAGTT-3’	5’-GCAATGGCCTTATCTGTGAC-3’
OPN	5′-ACTCGAACGACTCTGATGATGT-3′	5′-GTCAGGTCTGCGAAACTTCTTA-3′
OPG	5’-AAGGAGCTGCAGTACGTCAA-3’	5’-CTGCTCGAAGGTGAGGTTAG-3’
Caveol1	5’-AACAACCCGAACATCTACAACGGG-3′	5’-AAGGACTAACTCTAAGTCACGTAGTC

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
