# Peer review of "Bioengineered Fluorescent Nanoprobe Conjugates for Tracking Human Bone Cells: In Vitro Biocompatibility Analysis"

_materials, 2021, doi:10.3390/ma14164422_

Round 1

Reviewer 1 Report

The author validated that novel functionalized hybrid semiconductor bioconjugates made of a fluorescent quantum dot with the surface capped by chitosan, chemically modified with O-phospho-L-serine, are biocompatible and maintain the cellular behavior with different human cell sources. Overall, this author has shown many results, but it needs to be added because the experimental results are insufficient, especially the lack of discussion. The significant issues in the article are as the following.

Major concerns

Q1. In Figure 2, there is a difference between the size of the TEM images and the DLS result. Please check and correct the results.

Q2. In Figure 3, it is better to add cytotoxicity of QD as a control.

Q3. In Figure 4, it should be replaced the live&dead images with a higher resolution image.

Q4. In Figure 6b, there is no uptake data of HDF. It is better to add it.

Q5. In Figure 8,9, there is no FACS data of QD-Chi-LM and QD-Chi-HM. It is necessary to perform to ensure the cellular uptake in HDMSC and MG63.

Q6. According to the authors, LM was better in the results of the cellular uptake. However, why is the calcium deposit in HM (Figure 13)? The authors must clarify and provide evidence to prove this claim. 

Q7. Why are the markers measured on cells HBMSC and MG63 in Figure 14 different? The authors need to explain the reason.

Minor concerns

Q1. Overall, please change “in vivo” and “in vitro” to italics.

Q2. In Figure 7,8,9, it is better to unify group labeling.

Reviewer 2 Report

The manuscript “Bioengineered Fluorescent Nanoprobe Conjugates for Tracking Human Bone Cells: In vitro Biocompatibility Analysis” by Christiane Salgado et.al., presents bioengineered fluorescent CdS QDs (cadmium selenide quantum dots) stabilized by chitosan and bioconjugated to an OPS (O-Phospho-L-serine peptide) was used as a developed as an in vivo bioimaging tool. The concept and idea of this work is good, and it aligns well with the aims and scope of the journal. This manuscript can be accepted once the authors address the following suggestions. 1. Line 24, Abstract, ‘HBMSC’s’ Full form please 2. Line 39, “in vitro using soluble …..” . Kindly complete the sentence. 3. Kindly refer O-Phospho-L-serine as (OPS) abbreviation only . Kindly remove phosphoserine amino acids abbreviation as OPS. It is confusing. 4. Figure 1: Kindly improve and increase the font. Texts do not look clear/readable. Fig 1 (a):Inset, the x:asis is hidden with the figure legend. Kindly arrange the figures in a presentable and readable form. 5. Figure 2 : TEM figures look very hazy. I can understand that at 2.5-35 nm or below 5nm its difficult to get very perfect still figures. Kindly use a pointer/arrow to indicate that the black spots are the nanoparticles. All readers may not be able to understand this. Kindly improve and increase the font of all the figures in a presentable and readable form. Kindly work on the figure legends/numbers (a, b, c…etc.). they are not readable as well. 6. Figure 4: Scale Missing. Also, since the authors are using bone cells, staining with Alizarin Red on the 21-day and then acquiring the pictures would help confirm calcification of bone cells. Ideally on 7-day using cytoskeletal staining dye like Phalloidin would be good, 14-day RunX a pre-osteoblast/osteocyte/bone cell marker and 21-day alizarin red staining would be a good way to indicate progression/differentiation and verification that these were actually bone cells. If possible, I suggest the authors to kindly, see if this is doable. Pictures with MG63 is very hazy. Kindly improve the resolution. 7. Figure 5 and 6: Kindly improve and increase the font of all the figures in a presentable and readable form 8. Figure 10: I could not figure out whether the CdSQD particles were green or is it the cell cytoskeletal itself or is it that the CdSQD particles complete invaded the cellular structure and hence the entire cell itself appears to be fluorescent green. Again , if the developed nanoprobes are not cell/target/site specific and stains the entire cell this will ideally not be good fit for imaging? Kindly explain this. Staining entire cells can be done with cell staining biomarkers, we do not need nanoparticles for this. 9. Figure 13: kindly correct spelling and change from “Alizarim Red” to Alizarin Red 10. Kindly include a figure, chemical structure using Chem Draw or any other tool as per authors availability explaining the surface functionalized of the CdS QDS with chitosan, EDC-NHS-surface functionalization, and then addition/attachment or bioconjugation of OPS on the surface.

Round 2

Reviewer 1 Report

This manuscript has been appropriately revised according to the reviewers' comments. It reached a sufficient level to be published in this journal. But, should be aligned the scale bar in Figure 8. Please improve the resolution of all figures.

Author Response

Question 1: Aligned the scale bar in Figure 8. Improve the resolution of all figures.

Response1: Thanks for your comment. We changed and allined all the scalle bars in figure 8 and improved the resolution of all figures (Figures 1, 2, 4, 8 and 11 was improved using CorelDraw software). However, the journal's authors guidelines explain that is mandatory for publication that the images must have 300 DPI, that should decrease the quality of all figures.

Reviewer 2 Report

N/A

Author Response

Question 1: English language and style are fine/minor spell check required

Response 1: Thanks for your comment, the manuscript was totally reviewed and removed all minor spell errors according to the reviewer suggestion.

Question 2: Methods, results and conclusion can be improved.

Response 2: All the sections of the manuscript were reviewed in detail and improved according to the reviewer suggestion.